# User-Friendly Replication-Competent MAdV-1 Vector System with a Cloning Capacity of 3.3 Kilobases

**DOI:** 10.3390/v16050761

**Published:** 2024-05-11

**Authors:** Zhichao Zhang, Xiaojuan Guo, Wenzhe Hou, Xiaohui Zou, Yongjin Wang, Shuqing Liu, Zhuozhuang Lu

**Affiliations:** 1School of Public Health, Baotou Medical College, Inner Mongolia University of Science and Technology, Baotou 014040, China; 2NHC Key Laboratory of Medical Virology and Viral Diseases, National Institute for Viral Disease Control and Prevention, Chinese Center for Disease Control and Prevention, Beijing 100052, China

**Keywords:** mouse adenovirus 1, replication competent, vector, packaging capacity, transgene, infectious plasmid, Gibson assembly

## Abstract

Mouse adenoviruses (MAdV) play important roles in studying host–adenovirus interaction. However, easy-to-use reverse genetics systems are still lacking for MAdV. An infectious plasmid pKRMAV1 was constructed by ligating genomic DNA of wild-type MAdV-1 with a PCR product containing a plasmid backbone through Gibson assembly. A fragment was excised from pKRMAV1 by restriction digestion and used to generate intermediate plasmid pKMAV1-ER, which contained E3, fiber, E4, and E1 regions of MAdV-1. CMV promoter-controlled GFP expression cassette was inserted downstream of the pIX gene in pKMAV1-ER and then transferred to pKRMAV1 to generate adenoviral plasmid pKMAV1-IXCG. Replacement of transgene could be conveniently carried out between dual BstZ17I sites in pKMAV1-IXCG by restriction-assembly, and a series of adenoviral plasmids were generated. Recombinant viruses were rescued after transfecting linearized adenoviral plasmids to mouse NIH/3T3 cells. MAdV-1 viruses carrying GFP or firefly luciferase genes were characterized in gene transduction, plaque-forming, and replication in vitro or in vivo by observing the expression of reporter genes. The results indicated that replication-competent vectors presented relevant properties of wild-type MAdV-1 very well. By constructing viruses bearing exogenous fragments with increasing size, it was found that MAdV-1 could tolerate an insertion up to 3.3 kb. Collectively, a replication-competent MAdV-1 vector system was established, which simplified procedures for the change of transgene or modification of E1, fiber, E3, or E4 genes.

## 1. Introduction

Adenoviruses are a group of non-enveloped, icosahedral viruses with a genome of linear, double-stranded DNA of 26–48 kb in length [1]. Adenoviral vectors have been widely used in basic biomedical research, gene therapy, and vaccine development due to their properties of manipulable genome, high production, physicochemical stability, and efficient transient transduction [2,3,4,5]. However, the application of common adenoviral vectors is hampered by the pre-existing immunity in humans [6,7]. Interest has been attracted to constructing vectors based on rare serotype human adenoviruses (HAdV) or adenoviruses from other mammals or even birds [8,9,10].

From the angle of vaccine effectiveness, replication-competent adenoviruses have advantages over replication-defective ones. Replicating viruses can provide antigens in a higher amount and in a longer duration. Wild-type HAdV-4 and HAdV-7 have been prepared in the enteric-coated formulation and used as an oral vaccine against acute recruit respiratory syndrome by oral administration since the 1970s [11]. Another successful example is the rabies vaccine based on replication-competent HAdV-5. The gene encoding the G protein of rabies virus was inserted into the E3 region of HAdV-5 without an extra exogenous promoter; the prepared virus was mixed in baits, distributed in rural West Virginia by fixed-wing aircraft and could effectively immunize wild raccoons, gray foxes, and striped skunks [12,13].

Mouse adenovirus 1 (MAdV-1), historically called mouse adenovirus strain FL, was isolated by Hartley and Rowe while attempting to establish the Friend leukemia virus in mouse tissue culture [14]. Although MAdV-1 has a tropism for endothelial cells and cells of the monocyte/macrophage lineage, it can infect tissues throughout the mouse [15]. Recently, the receptors for MAdV-1 and MAdV-3 were found to be integrins αvβ6 and αvβ8, with which the RGD-motif in the fiber knob domain interacted [16,17]. Long-term co-evolution with the host finally leads to the host species-specificity of adenovirus, which limits the use of small animal models for the study of human-HAdV pathogenesis because HAdV cannot productively infect those animals. Mouse is the most commonly used laboratory animal, and the MAdV-mouse host system enables studies of viral pathogenesis in a natural host [15]. However, an easy-to-use reverse genetics system is still lacking for MAdVs.

The reverse genetics system of MAdV is many years behind that of HAdV in the use of new molecular biology technologies. To improve the rescue of recombinant virus, genomic DNA with covalently linked terminal protein (TP) was prepared from purified wild-type MAdV-1, digested with a unique restriction enzyme, mixed with a plasmid carrying mutation region and used to transfect mouse cells to rescue the mutant after homologous recombination. Plaque purification is often necessary to remove the contamination of wild-type MAdV-1 [18]. An infectious plasmid, which carried full-length genomic DNA, was constructed and used to generate mutants in 1999, although the low efficiency in virus rescue has limited its utility [19]. Recently, the method of homologous recombination in bacteria cells was used to construct adenoviral plasmids for MAdV-1 mutants or vectors [16,20]. As mentioned above, it seems that the efficiency of virus rescue was relatively low [16].

We previously simplified the procedure to construct adenoviral plasmid by combining the methods of restriction enzyme digestion and Gibson assembly (restriction-assembly), while the vector system was established by a strategy based on intermediate plasmids [21]. Briefly, an infectious plasmid was first constructed by one step of fusing a PCR product of the plasmid backbone to the adenovirus genomic DNA by Gibson assembly. A small intermediate plasmid was then separated from the infectious plasmid and used for target gene modification or exogenous gene insertion, and the modified fragment was brought back to the infectious plasmid to generate the final adenoviral plasmid by restriction-assembly. After that, the adenoviral plasmid was linearized by restriction digestion and transfected to packaging cells for virus rescue. The exchange of transgene was convenient, and it just needed one more step of restriction-assembly. Here, we attempted to establish an easy-to-use replication-competent MAdV-1 vector system by following this strategy.

## 2. Materials and Methods

### 2.1. Cells, Viruses and Mice

Mouse NIH/3T3 (ATCC CRL-1658), human HEp-2 (ATCC CCL-23), 293 (ATCC CRL-1573), A549 (CCL-185), and Hep-G2 (HB-8065) cells were cultivated in Dulbecco’s modified Eagle’s medium (DMEM) plus 10% fetal bovine serum (FBS; HyClone, Logan, UT, USA) at 37 °C in a humidified atmosphere supplemented with 5% CO_2_. Cells were detached by trypsin treatment, 1:3 split, seeded in new flasks, and would be used for virus infection when they reached 70–90% confluency the next day. Cells were regularly split twice a week. Wild-type MAdV-1 (ATCC VR-550) was amplified in NIH/3T3 cells. HAdV5-CG was an E1/E3-deleted HAdV-5 carrying CMV promoter-controlled GFP expression cassette and was constructed previously [22]. pKFAV4GFP was an adenoviral plasmid constructed previously and used as a PCR template here (https://www.nprc.org.cn/#/Adenovirus/homePage, accessed on 2 May 2024) [10].

Six- to eight-week-old female BALB/c mice were purchased from Beijing Vital River Laboratory Animal Technology (Beijing, China) and maintained under specific pathogen-free conditions. All experiments were carried out in strict compliance with the Guide for the Care and Use of Laboratory Animals of the People’s Republic of China and approved by the Committee on the Ethics of Animal Experiments of the Chinese Centre for Disease Control and Prevention.

### 2.2. Construction of Infectious Plasmid of MAdV-1

Primer sequences and PCR-related information were summarized in Appendix A. Overlap extension PCR was performed to amplify the plasmid backbone fragment containing the kanamycin-resistance gene and the origin of replication (Kan-Ori). The synonymous mutation was introduced to remove the RsrII site in the original template (Appendix A). Genomic DNA of wild-type MAdV-1 was extracted from virus-infected NIH/3T3 cells with the modified Hirt’s method [23]. PCR product was mixed with wild-type MAdV-1 genome (GenBank NC_000942), and Gibson assembly (NEBuilder HiFi DNA Assembly Master Mix, Cat. E2621, New England Biolabs, Ipswich, Massachusetts, USA) and bacterial transformation were performed to generate plasmid pKRMAV1. This plasmid was identified by restriction analysis and next-generation sequencing (Appendix A).

### 2.3. Construction of MAdV-1 Vectors

Plasmid pKRMAV1 was digested with EcoRI/RsrII, and the fragment of 11,213 bp was recovered from an agarose gel after electrophoresis and mixed with PCR products of 114 bp and 1579 bp (Appendix A). Gibson assembly was performed to generate an intermediate plasmid pKMAV1-ER. pKMAV1-ER was digested with SalI/PvuI, and the fragment of 10,306 bp was recovered and mixed with three PCR products of 2097 bp, 1323 bp, and 518 bp (Appendix A) for Gibson assembly to generate pKMAV1-ERCG. The EcoRI site inside the E1 region was removed by synonymous mutation in this step. pKMAV1-ERCG was digested with SwaI, pKRMAV1 was digested with EcoRI/RsrII, and the products were purified, mixed, and used to generate adenoviral plasmid pKMAV1-IXCG by DNA assembly (Figure 1). The luciferase-GFP fragment was amplified by PCR and integrated between the BstZ17I sites in pKMAV1-IXCG to generate pKMAV1-IXCLG by DNA assembly (Appendix A).

To load stuffer DNA fragments in various lengths downstream of the GFP expression cassette, a unique restriction site of SwaI was introduced in adenoviral plasmid pKMAV1-IXCG1K. Briefly, DNA fragments were amplified by PCR with the template of genomic DNA of 293 cells and used to construct a plasmid of pKan-stuf by DNA assembly. The plasmid of pKan-stuf contained a DNA fragment of 14.3 kb from the human X chromosome (GenBank NC_000023.11, from 152,882,412 bp to 152,896,668 bp) and was used as the template for amplifying fragments of stuffer DNA. G1K fragment was amplified by PCR and fused to CMVp-GFP PCR fragment; the product was used to replace CMVp-GFP in pKMAV1-IXCG to generate pKMAV1-IXCG1K by restriction-assembly (Appendix A). More stuffer DNA fragments (G2K, G3K, G4K, and G5K) were amplified and inserted into the SwaI site in pKMAV1-IXCG1K to generate adenoviral plasmids from pKMAV1-IXCG1K to pKMAV1-IXCG5K by restriction-assembly (Appendix A).

### 2.4. Rescue, Identification, Preparation and Titration of MAdV-1 Vectors

Flasks or plates were pre-treated with 0.1% gelatin (Cat. G9391, Sigma-Aldrich, St. Louis, MO, USA) in water at 37 °C for half an hour before being used to cultivate the packaging NIH/3T3 cells. NIH/3T3 cells were 1:3 split and would be ready for transfection when the culture reached 60–70% confluency the next day. Adenoviral plasmids were linearized by PmeI digestion and used to transfected to NIH/3T3 cells using jetPRIME reagent according to the manufacturer’s instructions (Cat. 114-15, Polyplus-transfection, Illkirch, France). The culture medium was removed and replaced with a maintenance medium of DMEM plus 2% FBS 6 h post-transfection. Half-medium change was performed every other day, and the culture system was maintained for 7–11 days until the occurrence of GFP foci. The cells and culture medium were harvested and subjected to three rounds of freeze-and-thaw and centrifugation. The supernatant was used as a seed virus for virus amplification in NIH/3T3 cells. The amplified virus was processed similarly, aliquoted, stored at −80 °C, and used for virus-infecting experiments.

To obtain purified virus genomic DNA, traditional CsCl density gradient ultracentrifugation was employed for virus purification. Briefly, viruses in a culture medium were precipitated by adding ammonium sulfate and then suspended in PBS [24]. Cell-associated viruses were released after three rounds of freeze-and-thaw and subjected to low-speed centrifugation to remove cellular debris. The harvested virus was purified by traditional ultracentrifugation [25]. It was found after titration assay that the purified virus lost 99% of the infection ability. The purified virus was lysed to release its genomic DNA in a buffer containing 10 mM EDTA, 0.1% SDS, and 0.2 mg/mL proteinase K (pH 7.6) at 50 °C for 2 h. Virus genomic DNA was recovered from the lysed purified virus (Genomic DNA Clean & Concentrator kit, Cat. D4010; Zymo Research, Irvine, CA, USA) and digested with restriction enzymes, followed by electrophoresis in agarose gel. SnapGene (www.snapgene.com, accessed on 2 May 2024) or pDRAW32 (www.acaclone.com, accessed on 2 May 2024) programs were used to draw plasmid maps or analyze restriction sites in plasmids or DNA fragments.

The infectious titers (IU/mL) of recombinant MAdV-1 viruses were determined with the limiting dilution assay on NIH/3T3 cells as described in detail elsewhere [26]. The infected cells (GFP+) in each well were counted under a fluorescence microscope, and titers were calculated. For wild-type MAdV-1, an immunofluorescence assay was employed to detect infected NIH/3T3 cells with antibodies against human adenovirus (Cat. 1401, ViroStat, Westbrook, ME, USA). HAdV5-CG was similarly titrated on human 293 cells.

### 2.5. Plaque Forming Experiment

Exponentially growing NIH/3T3 cells were 1:2.5 split and seeded on 6-well plates. Next, when cells reached a confluence of 90%, MAdV-1 viruses of 200, 500, or 1000 IU were diluted in 1 mL DMEM plus 2% FBS and added to the wells of the plate. After two hours of inoculation, viruses were removed by aspiration, and the cells were washed twice with 10 mM phosphate buffered saline (PBS) and covered with 2.5 mL DMEM containing 2% FBS and 1% low-melting agarose. Three days after infection, 2 mL fresh DMEM plus 2% FBS was supplemented to each well to replenish the nutrients. The liquid medium on the top was removed carefully, and 2.5 mL of 4% paraformaldehyde in PBS was added to each well 6 days post-infection. The cells were fixed at room temperature for 2 h, semisolid agarose gel was discarded, and cells were rinsed with running water and then merged in 1.5 mL crystal violet solution for half an hour. After removing the staining solution, wells were rinsed with running water several times and photographed with a cell culture scanner (Countstar Castor, Countstar, Shanghai, China). For the wells with 10–100 plaques, the number of plaques in each well was counted, and the area of each plaque was measured by using the Fiji image processing package [27]. The sizes of the plaques formed by different MAdV-1 vectors were compared with the Kruskal–Wallis nonparametric test.

### 2.6. One-Step Growth Curve

NIH/3T3 cells in 12-well plates with a confluency of 70% were infected with MAdV1-IXCG or MAdV-1 at an MOI of 2 IU/cell in 0.5 mL DMEM plus 2% FBS for 4 h. The virus-containing culture medium was aspirated, and the cells were washed twice with DMEM and cultured in 1 mL DMEM plus 2% FBS. At indicated time points, a culture medium of 600 µL in each well was transferred to a 1.5 mL tube and centrifugated at 300× *g* for 5 min. The supernatant of 400 µL was aspirated to a new tube without disturbing the precipitate. The cells in the wells were detached by scraping and aspiration. The detached cells, together with the culture medium in the well, were transferred to the tube that contained the remnant 200 µL medium, and all the tubes were temporarily stored at −80 °C before titration. After collecting all the samples, the tubes containing cells were subjected to three rounds of freeze-and-thaw and centrifugated at 5000× *g* for 5min, and the supernatant was titrated. The clarified medium (0.4 mL) was thawed and titrated directly. The number of viruses released to the medium (in infectious units, IU) was calculated by multiplying the titer value of the clarified medium with the volume of the culture medium (1 mL). The titer of the clarified medium was subtracted from that of the cell-medium mixture, and the answer times the volume of the sample (0.6 mL) was the number of cell-associated viruses in the well.

### 2.7. Gene Transduction of MAdV-1 Vector

Cells were seeded in 24-well plates 1 day before infection. MAdV1-IXCG virus was diluted in DMEM containing 2% FBS and added to the wells at a volume of 0.25 mL/well. The plate was placed on a mixer at 37 °C and rocked for 2 h with an oscillation frequency of 10 times per minute. The virus diluent was removed 4 h post-infection, and 0.5 mL fresh DMEM plus 2% FBS was supplemented to each well. Forty-eight or twenty-four hours post-infection, GFP expression was observed and photographed under a fluorescence microscope with a mounted CCD camera. After the observation, the cells were immediately detached by trypsin treatment, dispersed into single cells, suspended in PBS containing 1% FBS and 1.5% paraformaldehyde, and temporarily reserved in the dark at 4 °C. GFP expression in these fixed cells was assayed by flow cytometry within one week.

### 2.8. Replication of Viral Genome in Cells

MAdV1-IXCG was diluted in DMEM plus 2% FBS and was added to HEp-2 or NIH/3T3 cells in 12-well plates at an MOI of 0.5 IU/cell in 0.5 mL for 4 h. The medium containing viruses was discarded, and the cells were washed with DMEM and cultivated in 2 mL DMEM plus 2% FBS. At indicated time points, the media were discarded, and HEp-2 cells were collected after trypsin treatment, while NIH/3T3 cells were harvested directly by using a cell lifter. The genomic DNA from cells and viruses was extracted by using a genomic DNA extraction kit (Cat. no. DP304, TIANamp genomic DNA kit, TIANGEN, Beijing, China). TaqMan probe-based real-time PCR was performed to determine the copy number of the viral genome with the primers and probe targeting the GFP gene (Appendix A) under a condition of 95 °C 30 s, 95 °C 5 s, 60 °C 20 s, 45 cycles (Probe qPCR Mix, Cat. RR391A, TaKaRa, Dalian, China).

### 2.9. Bioluminescence Imaging

MAdV1-IXLG virus was diluted in PBS to a concentration of 3 × 10^5^ IU/mL. Before intranasal administration, mice were anesthetized by intraperitoneal injection of tribromoethanol at a dose of 400 mg/kg based on body weight, and then 40 μL virus diluent was dropped slowly and alternatively into both nostrils of each animal by using a pipette with a volume range of 10–100 μL. The dropping operation was performed with great care to avoid animal death caused by suffocation.

Before in vivo bioluminescence imaging, the mice were firstly wiped on the chest, abdomen, and hind limbs with cotton swabs dipped in depilatory cream; five minutes later, the covering hair was then removed by scrubbing the mice with cotton swabs dipped in water. D-luciferin in PBS was intraperitoneally injected into mice at a dose of 150 mg/kg (Cat. 40902ES02, Yeasen Biotechnology, Shanghai, China). Ten to 15 min later, animals were anesthetized in an anesthesia machine containing 2% isoflurane and imaged in ABLX6 with the isoflurane anesthesia gas on (Tanon, Shanghai, China). Regions of interest (ROI) were created on the image, and total flux was measured.

### 2.10. Statistical Analysis

The data are presented as the mean ± SD (standard derivation) and analyzed with one-way or two-way analysis of variance (ANOVA) unless otherwise indicated. *p* < 0.05 was considered statistically significant.

## 3. Results

### 3.1. Construction of Replication-Competent MAdV-1 Vector System

In the first place, an infectious plasmid was constructed for MAdV-1. A fragment containing a kanamycin-resistant gene and replication of origin for plasmid (Kan-Ori) was amplified by PCR. At both ends of Kan-Ori, there was a 34-nt region, which was identical to the distal end of the MAdV-1 genome, and a PmeI site flanking the 34-nt region. The 34-nt sequences and PmeI sites were included in the PCR product due to rational primer design. The Kan-Ori fragment was combined with MAdV-1 genomic DNA, and Gibson assembly was performed to generate the infectious plasmid pKRMAV1 (Appendix A).

An intermediate plasmid pKMAV1-ER was separated from pKRMAV1, which included E3, E4, E1, pIX, and partial IVa2 gene. Three polyA sites were found downstream of the pIX coding sequence. In pKMAV1-ER, a CMV promoter-controlled GFP expression cassette (CMVp-GFP) was inserted between the second and the third polyA sites to generate pKMAV1-ERCG by restriction-assembly. The modified fragment was restored to pKRMAV1 to generate the final adenoviral plasmid pKMAV1-IXCG. Notably, in the step of inserting CMVp-GFP, dual restriction sites of BstZ17I were intentionally added to facilitate the exchange of transgene expression cassette (Figure 1). For example, the CMV promoter, GFP coding sequence, and firefly luciferase coding sequence were amplified by PCR, respectively, and these three fragments were fused together by overlap extension PCR and then used to replace CMVp-GFP in pKMAV1-IXCG to generate pKMAV1-IXCLG (Appendix A).

Adenoviral plasmids pKMAV1-IXCG or pKMAV1-IXCLG were digested by PmeI to release the linear DNA containing the viral genome, which was used to transfect mouse NIH/3T3 cells. Recombinant viruses of MAdV1-IXCG and MAdV1-IXCLG were successfully rescued (Figure 2A and Appendix A). The MAdV1-IXCG virus was identified by restriction analysis of its genomic DNA (Figure 2B).

### 3.2. Transduction Ability of MAdV1-IXCG in Cell Lines

MAdV-1 has a tropism to mouse endothelial cells, and the cellular receptor for its fiber knob were found to be integrins αvβ6 and αvβ8. We tested its transduction to several cell lines (Figure 3). E1/E3-deleted HAdV-5 carrying CMVp-GFP (HAdV5-CG) was used as a control. HAdV5-CG could not transduce NIH/3T3, which did not express CAR, the receptor for HAdV-5. The most sensitive human cell line for MAdV1-IXCG was HEp-2, followed by Hep-G2 and 293. The transduction efficiencies were 62.1% or 92.4% when MAdV1-IXCG was used to transduce HEp-2 cells at MOIs of 0.1 or 0.2 IU/cell, respectively. More than 60% 293 or 80% Hep-G2 cells were transduced when MAdV1-IXCG was used at an MOI of 1 IU/cell (Figure 3). Because progeny viruses have been released from some MAdV1-IXCG-infected NIH/3T3 cells 48 h post-infection and the percentage of GFP+ cells detected at 48 h did not represent the real transduction rate, the experiment was repeated in NIH/3T3 cells, and the flow cytometry assay was conducted at 24 h post-infection. It was seen that about 67% of NIH/3T3 cells expressed GFP 24 h after being infected by MAdV1-IXCG at an MOI of 1 IU/cell (Appendix A).

### 3.3. Replication of MAdV1-IXCG in NIH/3T3 Cells

The plaque-forming experiment and one-step growth curve were used to evaluate the replication of MAdV1-IXCG in NIH/3T3 cells. It could be seen that the sizes of plaque formed by MAdV1-IXCG or wild-type MAdV-1 were not significantly different (Figure 4A,B). The cells were infected by MAdV1-IXCG or wild-type MAdV-1 at an MOI of 2 IU/cell, and one-step growth curves were drawn (Figure 4C). For both viruses, virus yields almost reached the peak 48 h post-infection, and the number of viruses released to the culture medium started to overpass that associated with cells 48 h post-infection. Another interesting phenomenon was also seen: the replication of MAdV-1 viruses could not be finely synchronized when infecting NIH/3T3 cells, the replication and release of progeny viruses lasted several days, and no typical peak was seen on the curve. Some cells died due to virus replication, and other cells kept growing and took the death cells’ place, being infected and lysed by the progeny virus. This process was repeated, although the balance was tilted to cell death. Collectively, these data indicated that MAdV1-IXCG could replicate as efficiently as wild-type MAdV-1 in NIH/3T3 cells.

### 3.4. Replication of Viral Genome in Human Cell Line

The amplification of the MAdV1-IXCG genome was determined in mouse NIH/3T3 or human HEp-2 cells after being infected with an MOI of 0.5 IU/cell. The replication of the viral genome reached the peak in NIH/3T3 cells 24 h post-infection, and at this time, no obvious increase of viral genome was detected in HEp-2 cells. Significant amplification of the viral genome was observed in HEp-2 cells 48 h post-infection, and the copy number detected at 96 h was 100 times higher than that at 4 h (Figure 5A). However, replication of the viral genome did not lead to the production of viable virus in HEp-2 cells being infected at the MOI of 0.5 IU/cell. If HEp-2 cells were infected at an MOI of 5 IU/cell, some infectious progeny viruses were produced. However, the total yield of progeny viruses was still lower than the amount of seeded parental viruses (Figure 5B). These data suggested that MAdV1-IXCG could replicate its genome in human cells. However, the amplified viral genome could hardly be packaged to form infectious virions.

### 3.5. In Vivo Replication of MAdV1-IXCLG

MAdV1-IXCLG carried two reporter genes of GFP and firefly luciferase, and in vivo, bioluminescence imaging could be employed to dynamically observe the replication of the virus in a mouse model. The mice were intranasally administrated with MAdV1-IXCLG of 1.2 × 10^4^ IU. The activity of luciferase was detected in the upper respiratory tracts 1 day post-infection, and it gradually spread to the chest, mainly the lungs, based on the shape formed by the bioluminescence signal. The activity of luciferase reached the peak 4 days post-infection, then receded and faded out beyond 12 days post-infection (Figure 6A,B). The body weight was monitored every day. The trend in body weights was the opposite of that in luciferase activity. They decreased, hit the bottom on day 4, and gradually grew up after that. The peak of bioluminescence usually appears on day 1 after immunocompetent mice are infected with replication-defective adenovirus carrying luciferase gene. Therefore, our data confirmed the replication and dissemination of MAdV-1 in mice.

### 3.6. Explore the Limits of Cloning Capacity

Non-coding sequences of various lengths were added between GFP CDS and the following polyA signal in pKMAV1-IXCG to generate five more adenoviral plasmids (Figure 7 and Appendix A). PmeI linearized adenoviral plasmids were used to transfect NIH/3T3 cells, and the culture systems were maintained for 11 days to observe the appearance of foci formed by GFP-positive cells (Figure 8A). Small GFP foci were found on day 3 post-transfection for MAdV1-IXCG1K and MAdV1-IXCG2K, on day 5 for MAdV1-IXCG3K, and on day 9 for MAdV1-IXCG4K. No GFP focus was seen for MAdV1-IXCG5K before day 11, when the culture started to decay due to cellular aging. Plaque-forming experiments were performed on all the rescued viruses. It could be seen that the size of the plaque gradually became smaller as the inserting fragment increased in length. The median value of plaque size formed by MAdV1-IXCG4K was about three times smaller than that formed by MAdV1-IXCG (Figure 8B). The yields of progeny viruses also went down as the length of insertion DNA increased (Figure 8C).

To evaluate the genetic stability of recombinant MAdV-1 viruses, the rescued seed viruses were passaged in NIH/3T3 cells four more times. The progeny viruses were purified by CsCl ultracentrifugation, and the virus genomic DNA was analyzed by restriction analysis. The restriction map of digested MAdV1-IXCG2K was as expected (Appendix A). However, the fragments containing inserts from digested MAdV1-IXCG3K or MAdV1-IXCG4K DNA became shorter or were missing, suggesting the rearrangement of viral genomes (Appendix A). The fragment of CMVp-GFP was 1.3 kb in length. Therefore, the replication-competent MAdV-1 could stably carry a transgene up to 3.3 kb.

## 4. Discussion

Rodents of mice and rats make up approximately 95% of all laboratory animals, while mice are the most commonly used animal in biomedical research [28]. Human adenoviruses cannot productively infect mice due to the host species specificity of adenoviruses, which makes mouse adenoviruses take an important role in studying host–adenovirus interaction [15]. Some MAdV-1 mutants have been successfully constructed [16,18,20]. However, easy-to-use reverse genetics systems or vector systems for loading transgene are still lacking for MAdV-1, which might result from two facts.

First, transfection of mouse cell lines is relatively difficult. 293 cells, a cell line often used for packaging HAdVs, can be transfected with many methods, and the transfection efficiency can be as high as 90%. In our laboratory, transfection to NIH/3T3 was often lower than 30% in efficiency for linearized adenoviral plasmids. The low transfection efficiency did bother us in the rescue of MAdV1-IXCG at the beginning, and thus, we chose to make efforts to extend the lifespan of a healthy cell culture after transfection. Half-medium change is a standard method to cultivate primary hematopoietic cells, which helps maintain a stable environment for cell growth and avoid loss of suspension cells [29,30]. We transplant half-medium change to cultivate transfected NIH/3T3. In addition, we pre-treated flasks or plates with 0.1% gelatin before seeding NIH/3T3 for transfection, and such an operation could promote cell attachment to the culture surface. The NIH/3T3 culture could be well maintained for approximately 10 days without cell passaging after implementing these measures. Generally, tens of GFP foci could grow up from NIH/3T3 cells transfected with linearized adenoviral plasmid in a T25 flask. Bieri et al. developed another strategy to overcome the low transfection efficiency in mouse cells [16]. They transfected I-SceI-expressing mouse cell lines with circular MAdV-1 plasmids instead of linearized ones. The transfection efficiency could be higher because a circular plasmid was used. Linearization would occur inside the cells, which expressed homing endonuclease. The transfection step is easy to carry out. However, the plasmid construction is somewhat complicated, and these I-SceI-expressing cell lines are indispensable but commercially unavailable.

Second, the genome of MAdV-1 cannot tolerate strong modification. MAdV-1 has a genome of 30.9 kb in length and belongs to the smallest ones in the genus of Mastadenovirus. For a small genome, dispensable sequences could be short, dispersive, and hard to find. Let us take the E3 region as an example. To construct a replication-competent HAdV vector, the E3 region was often deleted and used for bearing transgene. However, E3 in MAdV-1 is very different from that in HAdV-5. MAdV-1 E3 is embedded in the cluster of late genes [31]. The signal peptide coding sequence of E3 proteins was located inside the pVIII gene, and the distance between the stop codon of pVIII and the start codon of downstream fiber is about 720 bp, the maximum space of E3 that can be theoretically deleted. It was reported that silencing the expression of E3 proteins by introducing termination codons did not affect the growth of MAdV-1 in cultured cells [32]. We inserted GFP CDS downstream of the E3 signal peptide, deleted the partial E3 sequence to avoid possible damage to pVIII and fiber, and attempted to construct a replication-competent MAdV-1. The recombinant virus could not be rescued (unpublished data). There were publications to place exogenous genes in the E1A [16] downstream of L3-23K [33] or downstream of pIX [16,20]. All these modifications were performed with great care, and the strategy of fusion expression was often employed.

pIX is one of the two intermediate genes of adenovirus and has a relatively independent expression cassette. In 2004, Le et al. fused GFP to the C-terminal of pIX for tracking HAdV-5 infection [34]. Recently, GFP or H, F proteins of Canine Distemper Virus were successfully ligated to pIX to generate recombinant MAdV-1 [16,20]. Previously, we connected the miniSOG (mini-singlet oxygen generator) tag with pIX to observe the assembly of HAdV-5 inside the nucleus with an electron microscope [35].

Downstream of pIX is an ideal site for insertion of exogenous gene because placing transgene here will not interrupt the transcription of any viral gene. By analyzing the sequence between pIX and another intermediate gene IVa2 in the MAdV-1 genome, four polyA signals were found, with three rightward and one leftward (Figure 7). We chose to insert an independent transgene expression cassette between the second and the third polyA signal. Therefore, the first and the second polyA signals would be preserved for the transcription of E1 and pIX genes, and the transgene could use the third, while the transcription of IVa2 will be kept intact. The successful rescue of recombinant MAdV-1 viruses indicated that such a design was reasonable and feasible.

Independent transgene expression cassette has some advantages over the strategy of fusing transgene to the pIX gene. Although fusion protein could be cleaved to form separate peptides by intrinsic proteinase or the mechanism of ribosome skipping duration translation, some extra amino acids would be added to pIX and exogenous protein. The native promoter of the pIX gene is not a strong one, and the fusion strategy could result in a relatively lower expression of transgene. In addition, an independent transgene expression cassette provides more options to control the expression of transgene because we can replace the promoter when needed.

The vector system of pKMAV1-IXCG is user-friendly. The transgene expression cassette can be removed by BstZ17I digestion, and a new exogenous sequence can be generated by PCR and then directly added by Gibson assembly. Further modification of this system could be made with little effort. For example, as shown in Figure 7, once a unique cutter site of SwaI was introduced downstream of GFP, the insertion of more sequences into this site could be carried out conveniently by one step of restriction-assembly.

Besides loading transgene downstream of pIX, modification of E3, fiber, E4, E1, or pIX also becomes easier for MAdV-1 by using this system. The above-mentioned genes are all included in the intermediate plasmid pKMAV1-ER or pKMAV1-ERCG. First, these genes could be modified by site-directed mutation in the intermediate plasmid. For this aim, overlap extension PCR combined with restriction-ligation cloning is a commonly used approach. After that, these modifications could be transferred from the modified intermediate plasmid to the MAdV-1 genome by restriction-assembly. Of note is that the existing intermediate plasmid already covers the regions generally used for adenoviral vector modification. Of course, it is still possible to modify genes outside these regions with this system, in which case it is needed to separate another intermediate plasmid from infectious MAdV-1 plasmid to contain the region of interest.

In this study, we tested the cloning capacity of the vector system. It is well-known that HAdV-5 virion can package up to 105% of the wild-type genome length [36]. Hexon is the major capsomere of viral capsid. MAdV-1 hexon is 908 aa in length, which is 43 times shorter than that of HAdV-5. It was reported that the corresponding deletions in MAdV-1 hexon are located in the l_1_ and l_2_ loops of the molecule that form the protruding hexon towers on the external surface of the virion [37,38], which suggested that the virion of MAdV-1 might pack a genome larger than 105% of that of the wild type in length. While MAdV-1 has a small wild-type genome of 30.9 kb in length, our experiments indicated that recombinant MAdV-1 could package a genome of up to 34.2 kb. Namely, the replication-competent MAdV-1 vector constructed here has a cloning capacity of 3.3 kb.

In conclusion, we established a user-friendly replication-competent MAdV-1 vector system by inserting a transgene expression cassette downstream of pIX, which had a packaging capacity of 3.3 kb. This system could benefit the reverse genetics of MAdV-1 and vaccine development.

## 5. Patents

A patent application has been filed for this work by the National Institute for Viral Disease Control and Prevention, Chinese Center for Disease Control and Prevention.

## Figures and Tables

**Figure 1 viruses-16-00761-f001:**
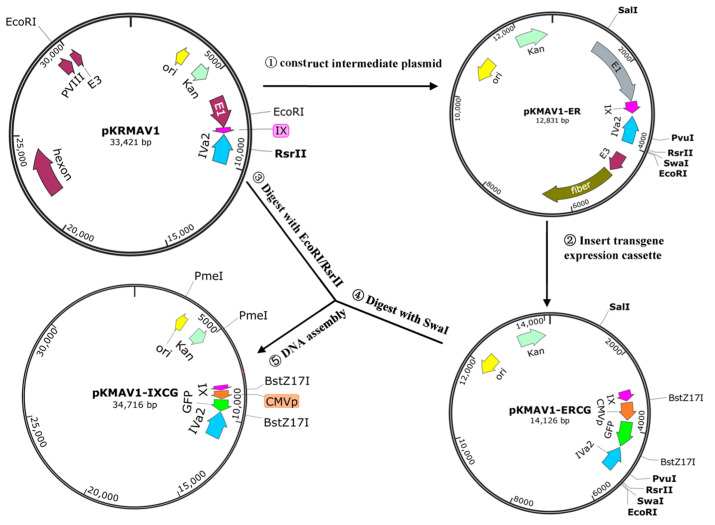
Schematic diagram of the construction of replication-competent MAdV-1 vector. An intermediate plasmid-based strategy was used to modify the genome of MAdV-1. pKRMAV1 was an infectious plasmid that contained the whole MAdV-1 genome. Intermediate plasmid pKMAV1-ER was generated by fusing the EcoRI/RsrII-digesting product of pKRMAV1 and other fragments of PCR products with Gibson assembly ①. pKMAV1-ER was a small plasmid with more unique restriction sites that could be used for site-directed mutation. CMV promoter-controlled GFP expression cassette was inserted downstream of the pIX gene to generate pKMAV1-ERCG ②. The modified intermediate plasmid of pKMAV1-ERCG was brought back to pKRMAV1 to generate the final adenoviral plasmid of pKMAV1-IXCG by restriction-assembly ③④⑤. The details can be found in the Materials and Methods section.

**Figure 2 viruses-16-00761-f002:**
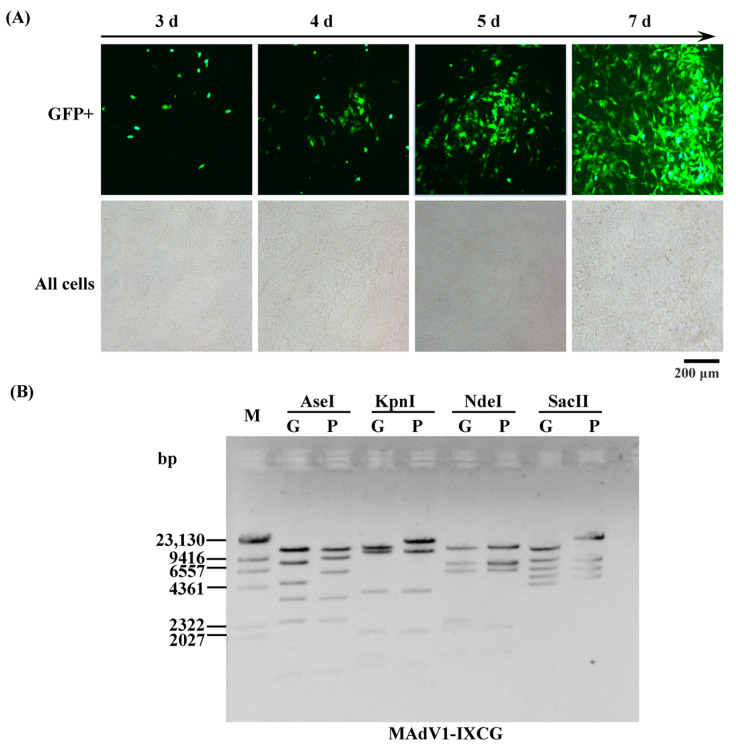
Rescue and identification of MAdV1-IXCG recombinant virus. (**A**) Rescue of MAdV1-IXCG virus. PmeI-linearized pKMAdV1-IXCG was used to transfect NIH/3T3 cells. The expression of GFP was observed under a fluorescence microscope. The occurrence and growth of GFP foci implied that the MAdV1-IXCG virus was successfully rescued. (**B**) Identification of MAdV1-IXCG by restriction analysis of its genomic DNA. Virus genomic DNA (G) was digested with the indicated restriction enzymes and resolved on 0.7% agarose gel by electrophoresis, and purified DNA of adenoviral plasmid pKMAdV1-IXCG (P) served as a control. The predicted molecular weights (bp) of digested fragments of MAdV1-IXCG genome were 1102, 2354, 3381, 4673, 8056, 12,382 for AseI; 1112, 1284, 1875, 3692, 10,735, 12,690 for KpnI; 1297, 1908, 2106, 2188, 5643, 6997, 11,866 for NdeI; and 3887, 4606, 5640, 7268, 10,844 for SacII. The predicted molecular weights (bp) of digested fragments of pKMAdV1-IXCG plasmid were 1102, 2354, 3381, 5818, 9382, 12,382 for AseI; 1112, 1875, 3692, 10,735, 16,445 for KpnI; 1297, 1908, 5643, 6765, 6997, 11,866 for NdeI; and 4606, 5640, 7268, 17,202 for SacII. Molecular weights of fragments less than 1000 bp were not given.

**Figure 3 viruses-16-00761-f003:**
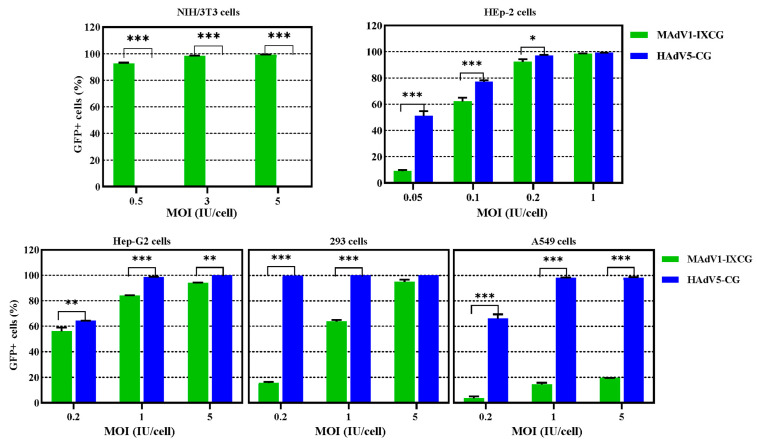
Transduction ability of replication-competent MAdV-1 vector carrying GFP reporter gene. Mouse NIH/3T3 and human adherent cell lines were infected with MAdV1-IXCG or control HAdV5-CG for 4 h at various MOIs; the cells were harvested 2 days post-infection, and the expression of GFP was assayed by flow cytometry. All the experiments were performed in duplicate, and the data shown are from one representative experiment out of the two performed. * *p* < 0.05, ** *p* < 0.01 and *** *p* < 0.001.

**Figure 4 viruses-16-00761-f004:**
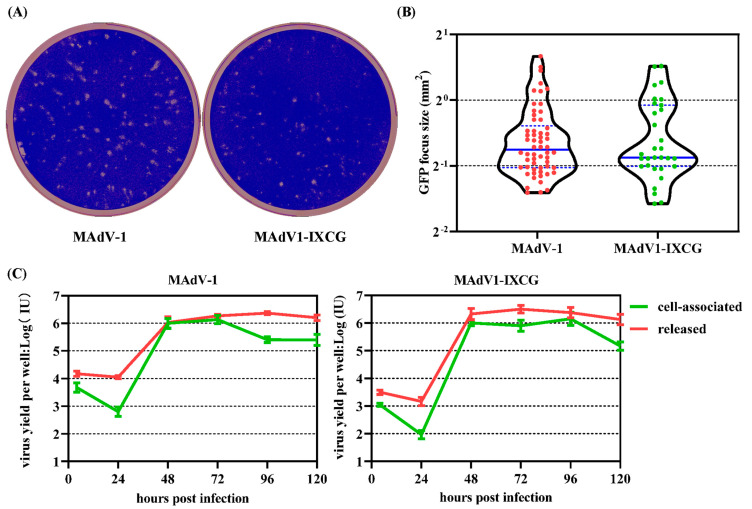
Growth of MAdV1-IXCG in packaging NIH/3T3 cells. (**A**) Plaque forming experiments. NIH3T3 cells in 6-well plates were infected with MAdV1-IXCG or the parental MAdV-1 at low doses for 2 h and then cultured in a semi-solid culture medium. The cells were fixed in paraformaldehyde, and the plaques were visualized by crystal violet staining 6 days post-infection. (**B**) The areas of total foci in one well were measured for each virus using the Fiji image processing package. The violin plots of the focus size data are shown. The size medians were compared by using the Kruskal–Wallis non-parameter test. (**C**) One-step growth curve. NIH/3T3 cells in 12-well plates were infected with MAdV1-IXCG or MAdV-1 at an MOI of 2 IU/cell for 4 h. Culture media and infected cells were harvested at indicated time points and used for titration. The yields of progeny viruses associated with cells or released to the culture medium were calculated, respectively.

**Figure 5 viruses-16-00761-f005:**
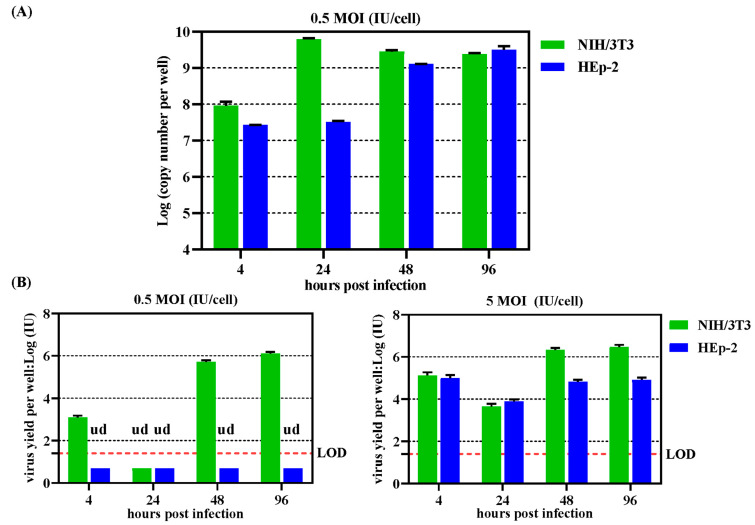
Replication of MAdV1-IXCG genome or infectious virus in human HEp-2 cells. (**A**) Mouse NIH3T3 or human HEp-2 cells in 12-well plates were infected in duplicate with MAdV1-IXCG at an MOI of 0.5 IU/cell for 4 h. Virus diluents were removed, and cells were washed twice before fresh maintenance media were added. Cells were collected at indicated time points, and total DNA was extracted. TaqMan probe-based real-time PCR was performed in triplicate to determine the copy number of the viral genome. The copy numbers of the viral genome in the well were calculated and shown. (**B**) NIH/3T3 or HEp-2 cells in 12-well plates were similarly infected with MAdV1-IXCG at MOIs of 0.5 or 5 IU/cell. At indicated time points, the cells, together with the culture medium, were harvested and used for titration. LOD—limit of detection; ud—undetected.

**Figure 6 viruses-16-00761-f006:**
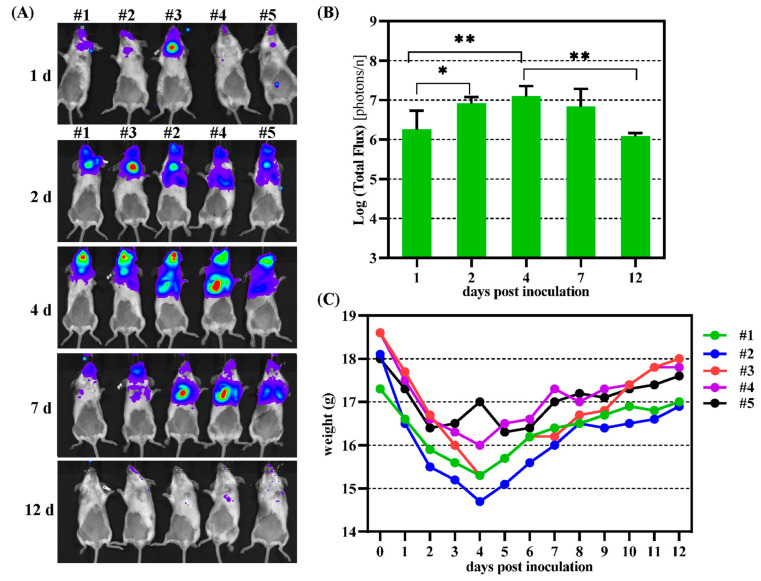
Biodistribution of MAdV1-IXCG in mice after intranasal administration. (**A**) The mice were intranasally infected with MAdV1-IXCLG at a dose of 1.2 × 10^4^ IU per mouse. In vivo bioluminescence imaging was performed to detect the activity of firefly luciferase in mice on sequential days post-virus inoculation. (**B**) Total photon flux measured after drawing the regions of interest (ROIs) in bioluminescence images. (**C**) Body weights of mice monitored on consecutive days post-infection. * *p* < 0.05, ** *p* < 0.01.

**Figure 7 viruses-16-00761-f007:**
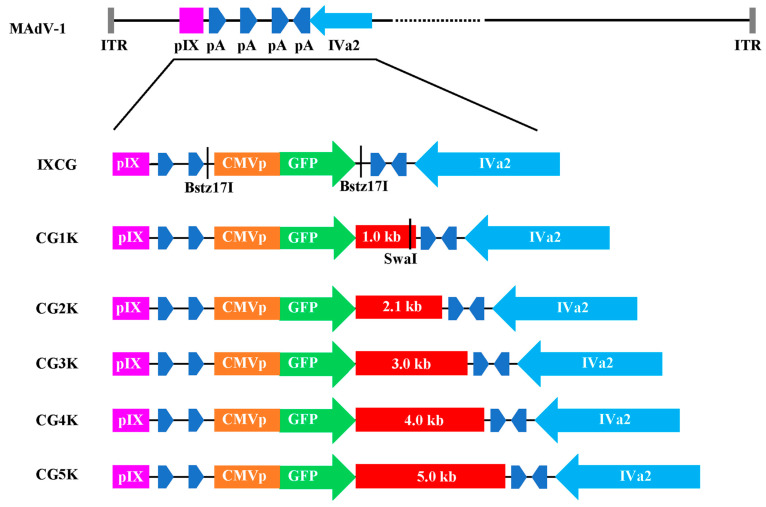
Schematic diagram of the transgene expression cassettes with gradually increasing size of stuffer DNA in recombinant MAdV-1 viruses. The details of the construction of these viruses were described in the Materials and Methods section. CMVp—CMV promoter; pA—polyA signal. IXCG, CG1K, CG2K, CG3K, CG4K, and CG5K were short names for MAdV1-IXCG, MAdV1-IXCG1K, MAdV1-IXCG2K, MAdV1-IXCG3K, MAdV1-IXCG4K, and MAdV1-IXCG5K.

**Figure 8 viruses-16-00761-f008:**
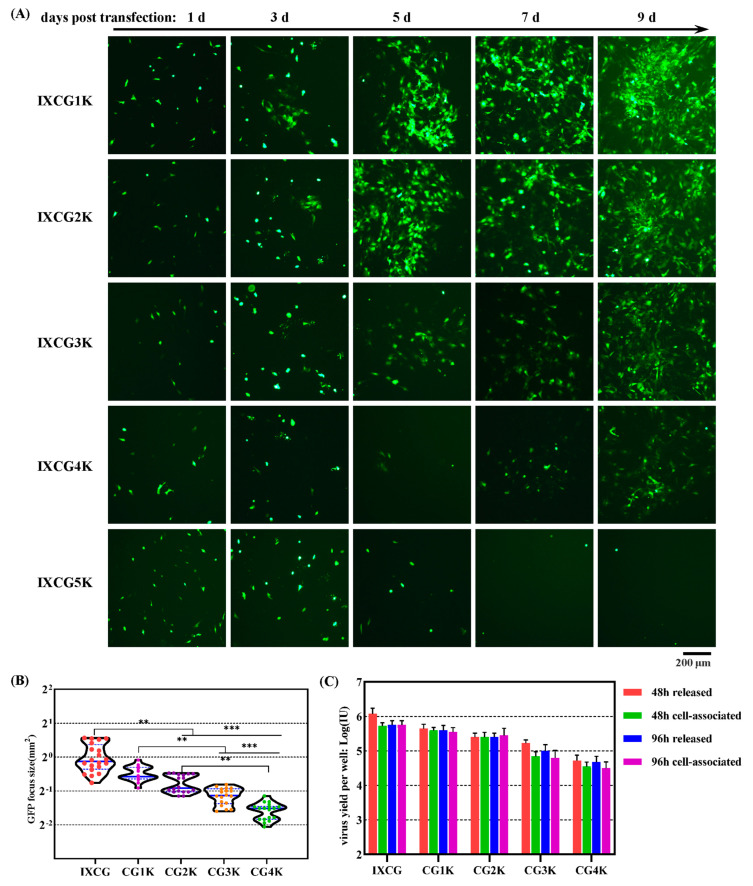
Growth of recombinant MAdV-1 viruses with increasing sizes of carried exogenous DNA. (**A**) Virus rescue in packaging cells. PmeI-linearized adenoviral plasmids were used to transfect NIH/3T3 cells. GFP-positive cells and occurrence of GFP foci were monitored by observing under a fluorescence microscope (MAdV1-IXCG5K failed in rescue). (**B**) Violin plots of the plaque size data. Rescued recombinant viruses and MAdV1-IXCG were subjected to plaque-forming experiments. The cells were fixed and stained 6 days post-infection. The areas of all plaques formed in one well were measured for each virus. The data were analyzed using the nonparametric Kruskal–Wallis test, and mean ranks were compared between two neighboring viruses. ** *p* < 0.01; *** *p* < 0.001. (**C**) Growth of recombinant viruses in fully infected packaging cells. NIH/3T3 cells in a 12-well plate were infectious with recombinant viruses at an MOI of 1 IU/cell for 4 h. The culture media and cells were harvested 48 or 96 h post-infection and used for titration. Virus yields were calculated. IXCG, CG1K, CG2K, CG3K, and CG4K were short names for MAdV1-IXCG, MAdV1-IXCG1K, MAdV1-IXCG2K, MAdV1-IXCG3K, and MAdV1-IXCG4K.

## Data Availability

Data are contained within the article and Appendix A.

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
