# Peer review of "User-Friendly Replication-Competent MAdV-1 Vector System with a Cloning Capacity of 3.3 Kilobases"

_viruses, 2024, doi:10.3390/v16050761_

Round 1
Reviewer 1 Report
Comments and Suggestions for Authors
In their manuscript titled "User-friendly replication-competent MAdV-1 vector system with a cloning capacity of 5 kilobases", Zhang et al. describe a novel, replication-competent MAV1 vector. The authors describe the method of generating the vector in intricate details, and provide data about its capacity to infect mouse and human cells in vitro and mice in vivo.
While the existence and potential availability of such vector is of interest, the data presented in the manuscript is better suited for another journal.
Comments on the Quality of English LanguageOnly minor editing is necessary.
Reviewer 2 Report
Comments and Suggestions for Authors
The authors construct and analyze an infectious clone of mouse Adenovirus i (MAV1). The vector expresses GFP or Firefly Luciferase and contains useful sites for the cloning of foreign genes of interest. The authors characterize in detail the properties of MAV1 generated from the infectious clone in different mouse and human cell lines in vitro and analyze infection properties in mice. Finally, the determine the maximum insert size the vector can accomodate and still maintain optimal infectious properties. Altogether, this is an excellent study that provides a valuable resource to the MAV1 field as well as a reagent that will be useful for investigators interested in expressing genes in mice.
Minor point: Table 1 and Fig 2B could be presented in a supplement, but this is not required.
Comments on the Quality of English LanguageMinor syntax corrections needed.
Reviewer 3 Report
Comments and Suggestions for Authors
The paper “User-friendly replication-competent MAdV-1 vector system 2 with a cloning capacity of 5 kilobases” by Zhang et al. report on a new innovative vector platform based on mouse adenovirus type 1(MAdV1). The authors report on the construction of new MAdV1 plasmid and present a new genetic work flow to modify this recombinant genome by Gibson assembly inserting transgenes a pre-determined site next to the pIX gene. These methods were described earlier for other adenovirus vector platform and this study is adapted them successfully to MAdV1. This study shows that inserting transgenes at the pIX locus did not disturb replication of MAdV1 in vitro and in vivo and also nicely shows the potential of the MAdV platform in cross-species settings in vitro using transduction of human cells as model. The authors also claim an unusually large cloning capacity for this vector platform. The study is interesting but needs to be completed at least regarding two crucial points:
Major points:
1. I am wondering how it is possible that MOI 0,2 of MadV1 based vector transduces more than 80% of NIH3T3 cells as shown in Figure 3? I guess it should not be more that 50%, since MOI 0,2 means that there was 1 infectious virus for 5 cells (20%)! The genome load needs to be determined for all preparations used in experiments throughout the paper, and the vector doses needs to be specified as vector genomes/cell (vg/c)! First, the TCID50 assays may underestimate the infectivity of the MAdV- vector preparations (the TCID50 on 293 cells is about 0,7 log lower for Ad5 than the pfu based assays). Second, the titer for Ad5-based vectors were, I guess, determined on 293 cells, which is a different target cell than NIH3T3. So, comparison of vector doses is fair only on the vg/c basis! This should not be a problem, since a well functioning real time PCR is presented in this paper for MAdV1.
2. The cloning capacity of MAdV1 based vector, determined in this study, is larger than I would expect for human adenovirus based vectors. Unfortunately, the results presented here are not sufficient to support this strong claim. I think, it would be essential to show that the larger genomes are stable after virus rescue. It is possible that some or most of the stuffer DNA is lost during vector propagation, which may or may not affect the GFP expression or progeny formation, which are tested here. The data presented in Figure 8B and C show that there is a substantial attenuation by the oversizing, which may induce instability selecting for genomes which loose inserted DNA. Moreover, in Figure 8 A, I do not see any evidence of spread of gfp expressing virus after 9 days! It would be important to show the restriction analysis of purified vector DNAs of the oversized constructs after stock preparation and compare their pattern to the original plasmids similar what was shown for the basic vector in Figure 2B to conclude the large cloning capacity of this vector platform.
I also found some minor issues, which should be addressed during the revision:
Minor point, corrections
1. Lane 13 replace “east-to-use” with “easy-to-use”
2. Table1 Please provide the source of pKFAdV4. You may consider moving this table to the supplementary material.
3. Lane 112: How the adenovirus DNA was purified?
4. Lanes 113-114. The supplementary figure S1 does not contain any data about pKRMAV1. Please insert the gel image for this plasmid as promised in the text.
5. Lane 114. If the construct was fully sequenced (sequence verified) as stated here, it would be beneficial, if the sequence or at least the NGS reads was (were) submitted to a public database according to the Viruses data availability policy.
6. lane 156 “The purified virus, which lost most of the infection ability,” how this was shown? Please insert data or, at least describe the assay, which you generated/used to draw this conclusion.
7. Lane 161 “The infectious titers (IU/ml) of adenoviruses were determined with the limiting dilu-161 tion assay on NIH/3T3” I guess this is only working for MadV1 derived viruses, can you describe how HAdV-5 derived viruses we were tested?
8. Figure 1 and all relevant figures later: Please specify which program did you used to generate the plasmid images and the fragment sizes you describe in figure legends.
9. Lane 257 replace “fly luciferase” with “firefly luciferase”
10. Lane 151 replace “To ob1tain” with “to obtain”
11. Lane 218 “TaqMan probe-based real-time PCR was performed”, Plese provide the PCR conditions for this assay.
12. Lane 245 : “and a PmeI site followed the 34-nt region” I guess the correct description would be “and a PmeI site flanking the 34-nt ITR region”.
13. Lane 248: “pKRMAV1 (Figure S1).” pKRMAV1 is not shown in Figure S1!
14. Lanes 279-280: replace “Virus genomic DNA was digested with restriction enzymes 279 and resolved on 0.7% agarose gel by electrophoresis, and adenoviral plasmid pKMAdV1-IXCG 280 served as a control.” with “Virus genomic DNA (G) was digested with the indicated restriction enzymes and resolved on 0.7% agarose gel by electrophoresis, and purified DNA of the adenoviral plasmid pKMAdV1-IXCG (P) served as a control.”
15. Lane 311: Replace “Wild-type” to “wild type”.
16: Lanes 313-314: “For both viruses, progeny viruses started to be detected 24 hours post infection” This cannot be concluded based on the data shown in Figure 4C, since there is no control shown for the remaining virus load used for the infection. There is clearly a progeny production between 24 and 48h post-infection and the titers measured for the wild type and the vectored MAdV1 are not significantly different.
17. Lane 367: Replace “replication and diffusion of MAdV-1 in mice” with replication and dissemination of MAdV-1 in mice".
Round 2
Reviewer 1 Report
Comments and Suggestions for Authors
The data presented in the manuscript are up to scientific standards. My objection to its suitability to Viruses still stands.
Reviewer 3 Report
Comments and Suggestions for Authors
I thank the authors for the careful revision and the very cooperative attitude dealing with my points. Indeed, the authors fulfilled all my queries when it was possible. I agree that vector genome determination as virus load measure needs highly purified samples and since the state-of-the-art purification did not function for the presented MAV vectors, I accept the second best method to determine virus load. Especially, since the authors dealt carefully with the comparison to the control vector now, the issue is fairly solved. The paper is now technically sound and contains all substantial information. I think, the paper now fulfils the standards, which are required for review publications in Viruses, therefore I suggested acceptance.